# Treatment Modalities of Recurrent Oral Mucosal Melanoma In Situ

**DOI:** 10.3390/medicina57090965

**Published:** 2021-09-14

**Authors:** Philipp Becker, Andreas Pabst, Monika Bjelopavlovic, Daniel Müller, Peer W. Kämmerer

**Affiliations:** 1Department of Oral and Maxillofacial Surgery, Federal Armed Forces Hospital, Rübenacherstr. 170, 56072 Koblenz, Germany; becker-ph@web.de (P.B.); andipabst@me.com (A.P.); 2Department of Oral and Maxillofacial Surgery—Plastic Operations, University Medical Center Mainz, Augustusplatz 2, 55131 Mainz, Germany; daniel.mueller@unimedizin-mainz.de; 3Department of Prosthodontics and Materials Science, University Medical Center Mainz, Augustusplatz 2, 55131 Mainz, Germany; monika.bjelopavlovic@unimedizin-mainz.de

**Keywords:** melanoma, in situ, oral neoplasm, head and neck cancer, hard palate, dental implant, irradiation, oral mucosa

## Abstract

Oral mucosal melanoma (OMM) represents an extremely rare entity that is associated with a poor prognosis due to late diagnosis and early metastasis. Here, extensive surgical therapy is the therapy of choice. In contrary, for OMM in situ, the respective therapeutical recommendations are lacking. In this case report, treatment modalities of an OMM in situ of the palate, including the maxillary alveolar process, are reported. The tumor relapsed twice despite adequate surgical therapy and reconstruction. Therefore, irradiation was performed as an adjuvant therapy. At a follow-up of two years, the patient was free from recurrences.

## 1. Introduction

Primary oral mucosal melanoma makes up less than 1% of all malignant melanomas and far below 1% of all oral malignant tumors. The main localization is the hard palate and the gingiva of the upper jaw [1]. The tumor originates histologically from oral melanocytes, which seem to be part of the oral immune system [2,3]. Oral melanoma, like other mucosal melanomas, normally has a bad prognosis with a high risk of recurrences [4]. Carcinogenic noxae, such as tobacco, but also various genetic mutations, are suspected to be risk factors for the development of oral mucosal melanoma (OMM) [1,5]. Dentists are often first to discover an OMM and can already make a clinical diagnosis if the OMM appears as a typical brown-to-blackish-pigmented nodule or macula [6]. A sample, or rather an excisional biopsy, should be taken as soon as possible to confirm the diagnosis. This is followed by staging examinations, which depend on the respective tumor stage. Radical surgery still represents the therapeutical “gold standard”. Depending on the stage, neck dissection and non-surgical treatment options, such as radiotherapy and medical therapy (e.g., checkpoint inhibitors), are also available [7]. Despite incremental therapeutical advances and sufficient therapeutic possibilities, OMM is associated with a limited and poor prognosis, which means that the earliest possible diagnosis and adequate therapy are crucial [6]. As opposed to invasive OMM, where resection margins of 2 cm are generally proposed, no evidence is available concerning the extension of sufficient resection margins for OMM in situ that is representing a clinical challenge. The following case report clearly shows how aggressive even an OMM in situ can appear, and shows a potential therapeutic algorithm.

## 2. Case Report

A 46-year-old male patient presented due to a rapidly progressive, blackish, irregular macula of the oral mucosa, measuring approximately 2 cm × 1.5 cm (Figure 1).

Eight years ago, a superficial, spreading, malignant melanoma of the left thigh was excised. There were no other relevant diagnoses or risk factors. Magnetic resonance imaging (MRI) confirmed a tumor-suspected mass in the upper jaw in regions 11 to 21 (Figure 2). Metastasis-suspected abnormalities of the cervical lymph nodes could not be identified.

The histopathological examination of an incisional biopsy showed an oral mucosal melanoma (OMM) in situ. At that time, a complete excisional biopsy did not appear to be possible due to the tumor size, invasiveness, and the expected extension of the resection. After discussing the case at the interdisciplinary tumor board, the tumor was excised together with the surrounding bone and teeth 12, 11 and 21 that were in close contact with the lesion. The definitive histopathological examination showed subepidermal nests of cells with round oval nuclei, pigmentation, hyperchromasia, small fluctuations in nucleus size and prominent nucleoli. The described nests were positive for HMB45 and Melan-A with a proliferation index of 25% (Ki-67). No Melan-A/HMB45 gradient was detectable. Overall, the finding was consistent with OMM in situ, which had been excised with sufficient safety margins (0.5 mm in all directions) (Figure 3).

Three months later, another blackish change in the oral mucosa of approximately 2 × 2 mm in region 012 was noticeable (Figure 4).

The mass, including the adjacent parts of the alveolar ridge and the palatine bone, was resected once again. Histopathologically, a R0-resected recurrence (0.5 mm in all directions) of the previously known OMM in situ without bone infiltration was confirmed. After secondary granulation of the excision wound, two dental implants were inserted in regions 012 and 021 (Figure 5).

Six months later, another recurrence occurred in the area of the hard palate (Figure 6), which was again excised extensively in toto (once again 0.5 mm in all directions) (Figure 7).

The prosthetic restoration was carried out with a ceramic, fully veneered, implant-supported bridge (Figure 8).

After consultation of the interdisciplinary tumor board and careful education of the patient, the informed decision of adjuvant irradiation was made. In order to prevent the tongue and buccal mucosa from receiving undesirable consequences of irradiation, a patient-specific device combining a tongue-depressing radiation stent and buccal (mucosal) retractor was designed (using bite impression material, a digital scan of this impression and construction of the device using the CAD software Meshmixer (Figure 9a,b)).

Afterwards, the tumor bed was fractionally irradiated with a circular safety margin of 2 cm with a radiation dose of 57.5 gray. The patient presented at regular intervals for clinical and ultrasound follow-up examinations. After 2 years of follow-up, there was no evidence for tumor recurrence or secondary tumors (Figure 10).

## 3. Discussion

A bluish, brownish or black macular or nodular change in the oral mucous membrane in the sense of a melanotic OMM is usually impressive and easy to recognize. In contrast to this, up to 40% of OMMs are amelanotic and therefore show no typical pigmentation, which makes a clinical diagnosis significantly more difficult [6,8,9]. However, the pigmented variants are usually diagnosed late as the patient’s symptoms such as pain, bleeding or ulceration appear at an advanced stage [10]. To confirm the suspected clinical diagnosis, a representative tissue sample must be taken for histopathological examination. Microscopically, melanotic OMM can often be successfully diagnosed using hematoxylin and eosin staining. In addition, immunohistochemical markers, such as HMB45 and Melan-A, are available to confirm the diagnosis and to histologically differentiate the amelanotic OMM from lymphomas, undifferentiated carcinomas or sarcomas [11]. The staging examinations of OMM depend on the respective tumor stage. According to current German guidelines, a clinical examination is sufficient to stage melanoma in situ. In advanced stages, MRI of the head, whole-body cross-sectional imaging using PET-CT, CT or a skeletal scintigraphy and the determination of tumor markers S100B and LDH are added [7]. The therapy of choice and the only potentially curative treatment of an OMM is a radical surgical resection with a sufficient safety margin of usually 1–2 cm [12].

The expansion of the surgical therapy of OMM is controversially discussed. In this context, an extended surgical resection of OMM may result in an increased disease-free survival [13]. López et al. reported on a bad prognosis of primary mucosal melanoma with 5-year disease-free survival rates of less than 20% [14]. With a special focus on desmoplastic melanoma of the oral cavity, poor 5-year disease-free and overall survival rates (0%, 55%) were reported [15]. A review reported recurrences in 11.6% and mortality in 54.8% of cases of primary oral melanoma [16].

Surprisingly, there are several case reports of a melanoma in situ of the hard palate, some of which tend to recur even after years, despite adequate therapy [17]. In contrast, with invasive OMM, elective neck dissection can take place, since lymph node metastases can be detected in up to 75% of cases [18]. In addition, patients with invasive tumors or R1 situations can be offered adjuvant radiation therapy. In the case of metastatic OMM or inoperability, after the individual mutation status has been determined, targeted systemic therapy using BRAF, MEK, KIT or checkpoint inhibitors can be carried out in order to extend overall survival with an acceptable rate of side effects [7]. Overall, OMM has a poor prognosis [19].

## 4. Conclusions

This case represents a special surgical challenge since guidelines and recommendations concerning the surgical safety distance for OMM in situ are still missing. Therefore, the surgical extension as well as a potential radiation therapy are based on the local anatomical conditions, the assessment of the surgeon and the patient’s desire.

## Figures and Tables

**Figure 1 medicina-57-00965-f001:**
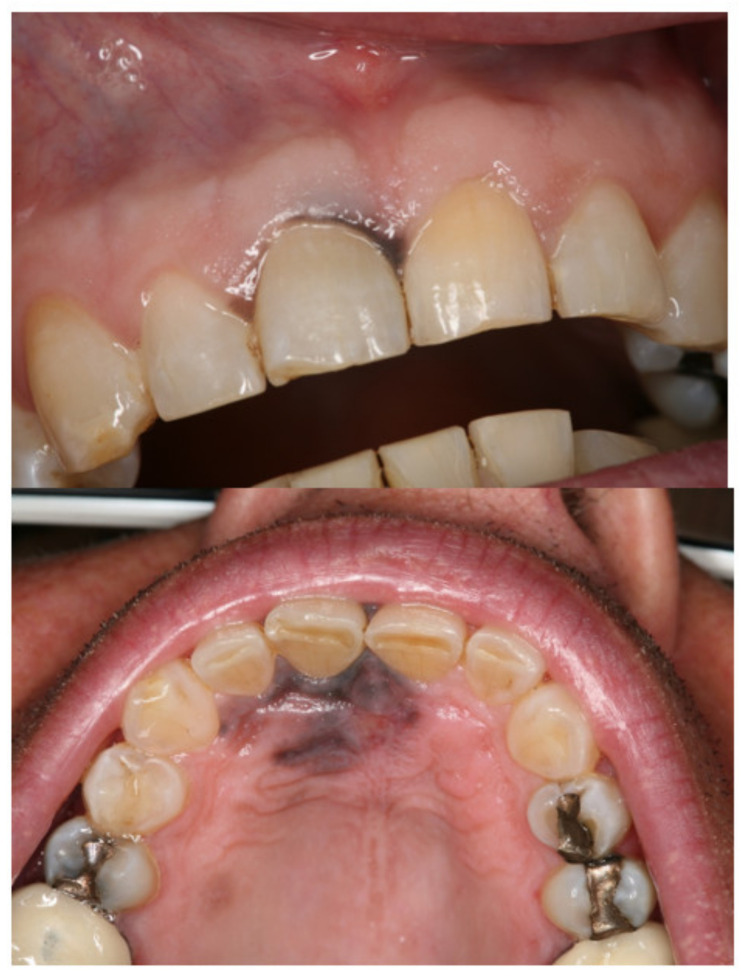
Clinical findings at first visit: on the labial marginal gingiva of tooth 11 and in the area of the anterior hard palate, an indistinctly delimited blackish oral mucosal change is evident, which is clinically highly suspect of an oral mucosal melanoma.

**Figure 2 medicina-57-00965-f002:**
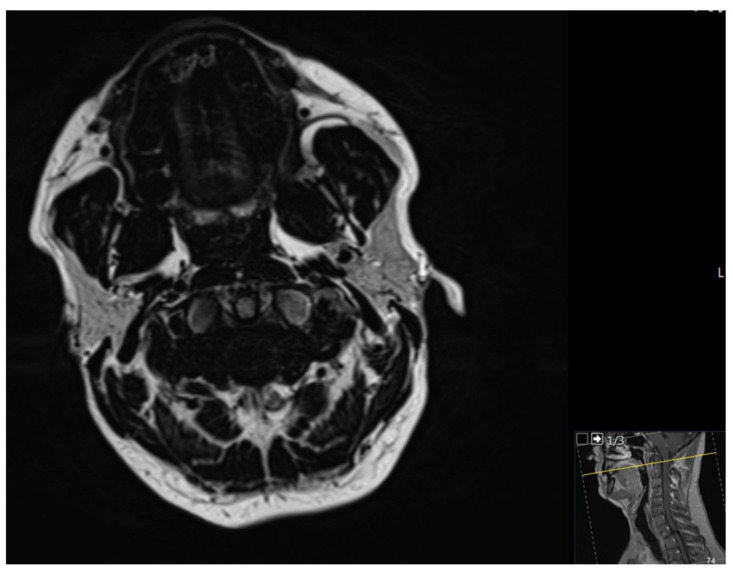
MRI (axial slices, T2)–tumor formation in the front region of the upper jaw.

**Figure 3 medicina-57-00965-f003:**
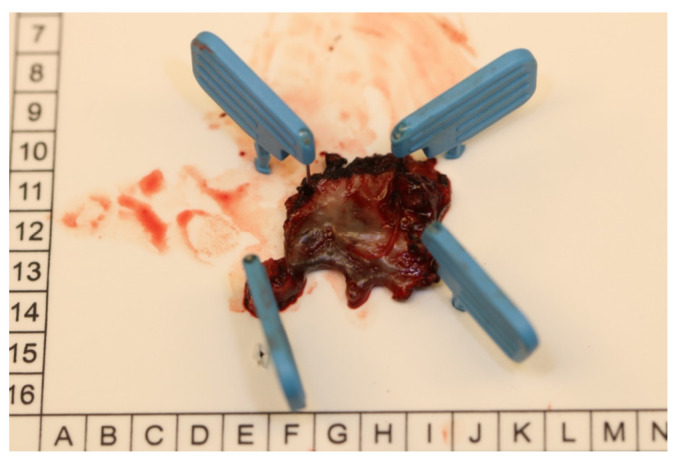
After resection, the specimen was oriented and fixed for histopathological examination.

**Figure 4 medicina-57-00965-f004:**
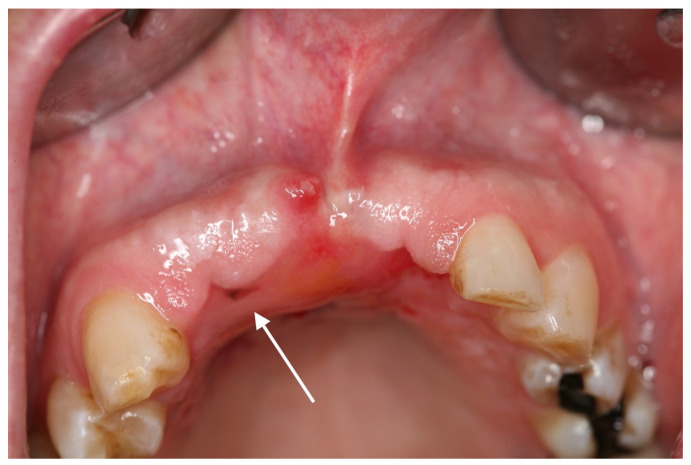
First recurrence of oral mucosal melanoma: in region 012, there was a 2 × 2 mm dark mucosal macule (white arrow), which was highly suspicious of a relapse of the OMM.

**Figure 5 medicina-57-00965-f005:**
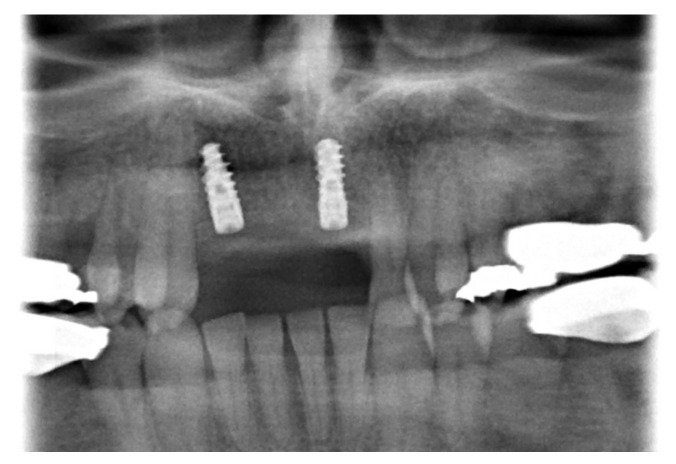
Radiographic follow-up after implant placement: two dental implants (each 3.7 × 10 mm, BLX, Straumann, Basel) were placed.

**Figure 6 medicina-57-00965-f006:**
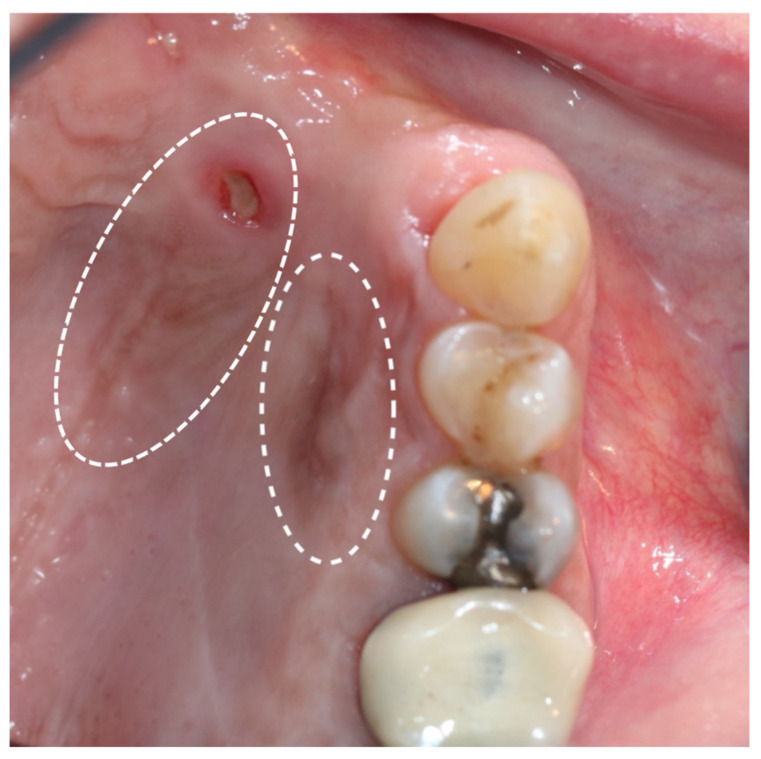
Second recurrence of OMM: irregular, light-brown changes and a fibrin-covered ulceration in the palatal mucosa are evident (white circles).

**Figure 7 medicina-57-00965-f007:**
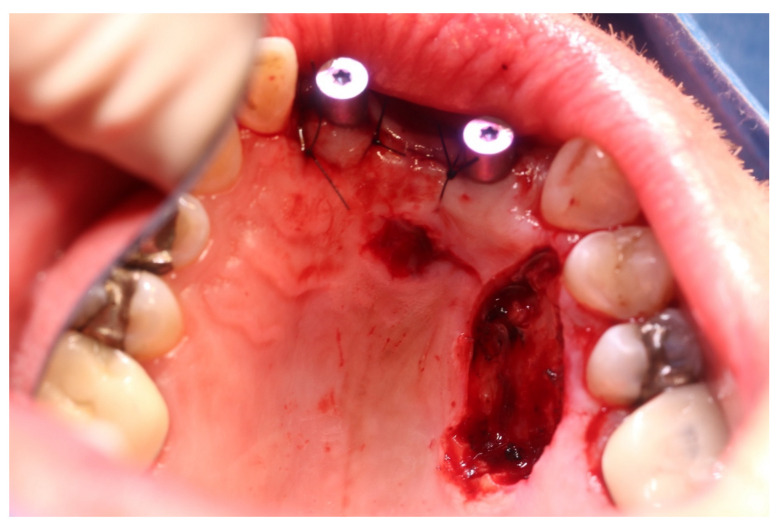
Condition after resection of the second recurrence and exposure of the implants: the changes in the oral mucosa were resected with safety margins. The implants in regions 12 and 21 were exposed and restored with gingival formers.

**Figure 8 medicina-57-00965-f008:**
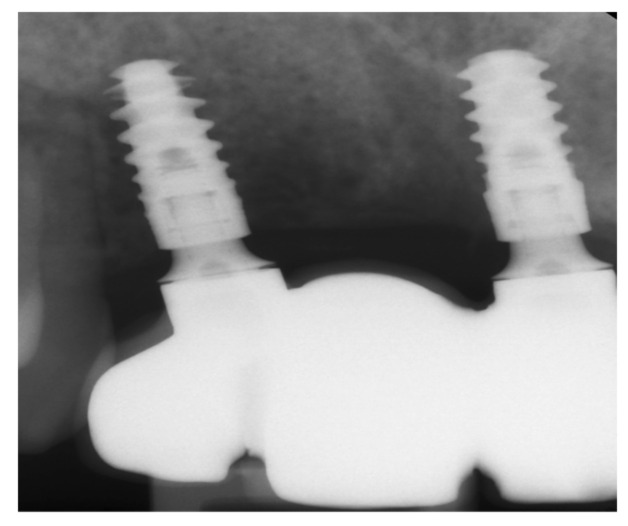
Single-tooth radiograph showing dental implants together with the final restauration.

**Figure 9 medicina-57-00965-f009:**
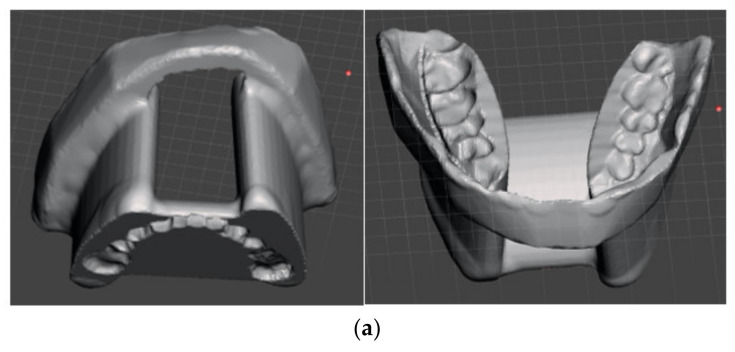
Patient-specific 3D-printed device combining a tongue-depressing radiation stent and buccal retractor: (**a**) CAD construction and (**b**) device installed in the patient.

**Figure 10 medicina-57-00965-f010:**
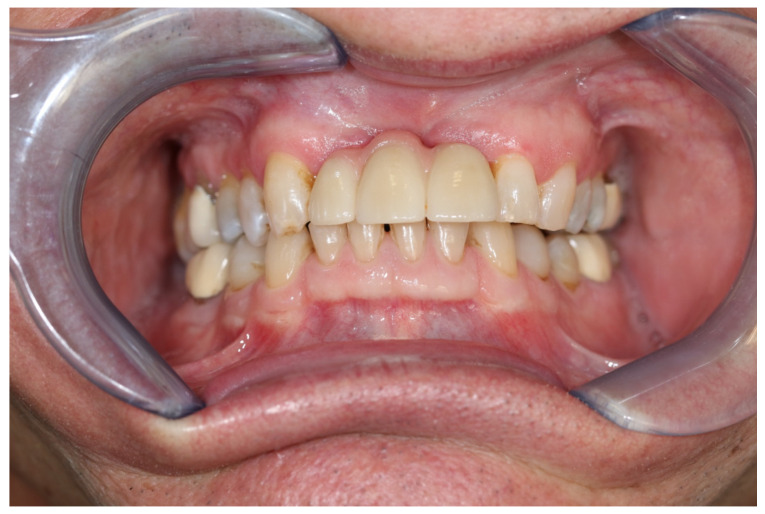
Oral situation after 2 years with prosthetic restauration.

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
