# Peer review of "Treatment Modalities of Recurrent Oral Mucosal Melanoma In Situ"

_medicina, 2021, doi:10.3390/medicina57090965_

Round 1
Reviewer 1 Report
An interesting case report about an oral melanoma in situ treated with surgery two times and then with radiotherapy after relapsing. The main limitation of the case is the follow-up. as relapse may happen various years after treatments.
only minor revisions:
page 1 line 27 you should add: "oral melanoma, like other mucosal melanomas, has normally a really bad prognosis with a high risk of recurrences" and cite an article such as: doi: 10.3390/medicina57040359.
The main problem with this paper in my opinion is in the discussion, where you state that melanoma in situ recurs only in 2% of cases. This may be the case of cutaneous melanomas, but it is hard to believe that this applies to oral and mucosal melanomas; the article you cite refers, as a matter of facts, to cutaneous melanoma; I suggest the authors only select articles that talk about mucosal melanomas, that have different and more aggressive behavior, and modify discussion section
Thank You
Author Response
- Concern of the reviewer: “Page 1 line 27 you should add: "oral melanoma, like other mucosal melanomas, has normally a really bad prognosis with a high risk of recurrences" and cite an article such as: doi: 10.3390/medicina57040359.”
Our response: We thank the reviewer for the review of the manuscript. We added the recommended statement and the reference.
Revised text: Oral melanoma, like other mucosal melanomas, has normally a bad prognosis with a high risk of recurrences [4].
Lombardo, N., M. Della Corte, C. Pelaia, G. Piazzetta, N. Lobello, E. Del Duca, L. Bennardo, and S. P. Nisticò. "Primary Mucosal Melanoma Presenting with a Unilateral Nasal Obstruction of the Left Inferior Turbinate." Medicina (Kaunas) 57, no. 4 (2021).
- Concern of the reviewer: “The main problem with this paper in my opinion is in the discussion, where you state that melanoma in situ recurs only in 2% of cases. This may be the case of cutaneous melanomas, but it is hard to believe that this applies to oral and mucosal melanomas; the article you cite refers, as a matter of facts, to cutaneous melanoma; I suggest the authors only select articles that talk about mucosal melanomas, that have different and more aggressive behavior, and modify discussion section.”
Our response: We modified the discussion section and selected references dealing with information about the recurrence rate of mucosal melanomas.
Revised text: The expansion of the surgical therapy of OMM is controversially discussed. In this context, an extended surgical resection of OMM may result in an increased disease-free survival [13]. López et al. reported about a bad prognosis of primary mucosal melanoma with 5-year disease-free survival rates of less than 20% [14]. With a special focus on desmoplastic melanoma of the oral cavity, poor 5-year disease-free and overall survival rates (0%, 55%) were reported [15]. A review reported about recurrences in 11.6% and mortality in 54.8% of the cases of primary oral melanoma [16].
- Hasan, S., S. F. Jamdar, J. Jangra, and S. M. Al Beaiji. "Oral Malignant Melanoma: An Aggressive Clinical Entity - Report of a Rare Case with Review of Literature." J Int Soc Prev Community Dent 6, no. 2 (2016): 176-81.
- López, F., J. P. Rodrigo, A. Cardesa, A. Triantafyllou, K. O. Devaney, W. M. Mendenhall, M. Haigentz, Jr., P. Strojan, P. K. Pellitteri, C. R. Bradford, A. R. Shaha, J. L. Hunt, R. de Bree, R. P. Takes, A. Rinaldo, and A. Ferlito. "Update on Primary Head and Neck Mucosal Melanoma." Head Neck 38, no. 1 (2016): 147-55.
- Min, S. K., J. H. Jeong, K. M. Ahn, C. W. Yoo, J. Y. Park, and S. W. Choi. "Desmoplastic Melanoma of the Oral Cavity: Diagnostic Pitfalls and Clinical Characteristics." J Korean Assoc Oral Maxillofac Surg 44, no. 2 (2018): 66-72.
- de Castro, M. S., B. S. A. Reis, D. A. Nogueira, M. L. de Carli, J. A. C. Hanemann, A. A. C. Pereira, O. P. Almeida, and F. F. Sperandio. "Primary Oral Melanoma: A Clinicopathologic Review and Case Presentation." Quintessence Int 48, no. 10 (2017): 815-27.
Reviewer 2 Report
The presented material is exceptionally interesting due to the proposed new directions in treatment of oral mucosal melanoma in situ. Regardless of the risk of reoccurrence, the proposed treatment option could be useful to oral pathology specialists.Author Response
Reviewer #2
“The presented material is exceptionally interesting due to the proposed new directions in treatment of oral mucosal melanoma in situ. Regardless of the risk of reoccurrence, the proposed treatment option could be useful to oral pathology specialists.”
Our response: We thank the reviewer for the review of the manuscript.
Revised text: Not required